Early detection of abiotic stress in plants through SNARE proteins using hybrid feature fusion model

T. Bhargavi
http://orcid.org/0000-0003-2920-4640 D. Sumathi sumathi.d@vitap.ac.in
School of Computer Science and Engineering, VIT-AP University , Amaravati, Andhra Pradesh , India
Balas Valentina Emilia
Electronic publication date: 2024 Aug 5
Publication date: 2024
Volume: 10
Electronic Location ID: e2149
Received 2023 Nov 6; Accepted 2024 May 31
Copyright: © 2024 T and D
Copyright year: 2024
Copyright holder: T and D
License: This is an open access article distributed under the terms of the Creative Commons Attribution License, which permits unrestricted use, distribution, reproduction and adaptation in any medium and for any purpose provided that it is properly attributed. For attribution, the original author(s), title, publication source (PeerJ Computer Science) and either DOI or URL of the article must be cited.
License URL: https://creativecommons.org/licenses/by/4.0/

Keywords: Abiotic stress, SNARE proteins, CNN, Bi-LSTM, Agriculture, Feature fusion, Deep learning

Funding: The authors received no funding for this work.

==============================
Agriculture is the main source of livelihood for most of the population across the globe. Plants are often considered life savers for humanity, having evolved complex adaptations to cope with adverse environmental conditions. Protecting agricultural produce from devastating conditions such as stress is essential for the sustainable development of the nation. Plants respond to various environmental stressors such as drought, salinity, heat, cold, etc. Abiotic stress can significantly impact crop yield and development posing a major threat to agriculture. SNARE proteins play a major role in pathological processes as they are vital proteins in the life sciences. These proteins act as key players in stress responses. Feature extraction is essential for visualizing the underlying structure of the SNARE proteins in analyzing the root cause of abiotic stress in plants. To address this issue, we developed a hybrid model to capture the hidden structures of the SNAREs. A feature fusion technique has been devised by combining the potential strengths of convolutional neural networks (CNN) with a high dimensional radial basis function (RBF) network. Additionally, we employ a bi-directional long short-term memory (Bi-LSTM) network to classify the presence of SNARE proteins. Our feature fusion model successfully identified abiotic stress in plants with an accuracy of 74.6%. When compared with various existing frameworks, our model demonstrates superior classification results.

Introduction

Over the past few years, agriculture has become increasingly vital in the global economy, playing a critical role in ensuring safe and efficient food production. This has led to a rising demand for smart farming techniques. The integration of computer vision and deep learning has enabled the use of advanced technologies in agricultural automation, significantly benefiting small-scale farming practices (Ünal, 2020). The Indian economy relies on agriculture by acquiring sustainability in agricultural production by including advanced technologies in farm management. Stress in plants is the main threat to productivity. Identifying stress at the initial stage can curb the loss because this is a very challenging task in agriculture (Chung, Breshears & Yoon, 2018). Abiotic stressors are environmental factors that negatively impact plant growth and development, leading to reduced yields below optimal levels. These stressors, which include factors like temperature extremes, water scarcity, and soil salinity, can significantly affect crops and commercial plants. In some cases, they can cause crop production to decrease by up to 70%, limiting plants to operate at just 30% of their genetic potential (Abbas & El-Manzalawy, 2020). Plant growth under unfavourable conditions is termed plant stress. Plant stress is categorized into biotic stress which is caused due to the invasion of bacteria, viruses, fungi, pathogens, and many more whereas abiotic stress occurs mainly due to the intervention of non-living organisms. stresses caused by heat, drought, and salinity (Cushman & Bohnert, 2000). In a few cases, these stressors acts as a defensive mechanism that protects the plant from deviations caused by external factors. The impact of stress shows its imprint in the decline of food production which further affects the supply-chain management of the agricultural industry (Jansen & Potters, 2017). Furthermore, fluctuations in climatic conditions can also lead to great damage (Rico-Chávez et al., 2022). Artificial Intelligence-based methodologies showed promising results in analyzing the mechanism of plant stress, thereby helping farmers monitor crop stress in the early stages (Fenu & Malloci, 2021). Abiotic stress greatly impacts the physical and biological status of the plant. The vital biochemical present in plants helps them to increase the quantity of the produce (Moghimi, Yang & Marchetto, 2018). Abiotic stress disturbs the gene structure of the plants which leads to the fatal destruction of the yield. Abiotic stress in plants is mainly contributed by different phases of plant growth, advancements in the genetic structure of the plants, and various biotic and abiotic stressors (Kazan, 2015). Proteins play an essential role in the biological functioning of plant tissue. Identifying protein-protein interaction in plants is vital to know the organ and tissue formation, cell structure, and plant defense mechanism (Yang et al., 2011). Therefore, analysis of the interaction between sequence-based proteins is important to identify the stress in plants (Zhang, Gao & Yuan, 2010; Khan & Kihara, 2016). Proteins are the most essential building blocks present in every life science. Understanding the underlying structure and functions of proteins is a challenging task. Protein sequences are composed of a series of alphabets and machine learning and deep learning methodologies are used to decode the sequence (Ofer, Brandes & Linial, 2021). Deep learning, a subset of Artificial Intelligence is extensively used to learn the data stored in multiple layers as sequences of hidden amino acid details from a protein sequence (LeCun, Bengio & Hinton, 2015). SNARE proteins (soluble N-ethylmaleimide-sensitive factor activating protein receptors) are vital structures that hold vital information about cell membrane formation (Kha, Ho & Le, 2022). By considering and analyzing the key importance of SNARE proteins in the transportation of essential information, several techniques have been used to decode the information. One of them is to investigate the SNARE from the unknown motif information with the help of bioinformatics (Kloepper et al., 2007). Furthermore, SNAREs are identified with the help of position position-specific scoring matrix and are fed into a 2D convolutional neural network in the form of images thereby extracting the vital information from the sequential data (Le & Nguyen, 2019). Numerous environmental stresses trigger the plant’s biological structure leading to changes in the life cycle such as changes in the functionality of the antioxidants involved, reconstruction of the endomembrane structure, accumulation of the visible solutes in the gene expression, and change in the transportation cycle of the protein substances (Wang et al., 2020). Nowadays, crop damage is mainly due to climatic changes and incorrect human actions resulting in the decline of food production (Mousavi-Derazmahalleh et al., 2019). Plants developed a resistant environment by inhibiting the nature of adapting to the stress as well as coping with uninvited changes in the environment made them benefit from the adversities created by the unfavorable nature leading them to build a strong defense against their constantly shifting environment (Kalinowska & Isono, 2018). Throughout their life cycles, plants are continuously exposed to several abiotic challenges, including biotic stresses and drought, excessive salinity, heat, cold, freezing, UV-B, and osmotic pressures (Sanderfoot, 2007). Proteins are a jack of all bio-molecular functions of a living cell and they are represented as a chain of patterns built with amino acid composition with unique lengths. Feature extraction is the main step that exposes the interaction and the involvement of amino acids in the function of the cell. Traditional methods extract features from the Basic Local Alignment Search Tool (BLAST) (Xu et al., 2021). To extract the descriptive features, evolutionary methods like Auto-Cross-Covariance (ACC) (Dong, Zhou & Guan, 2009) and Separated-Dimer (SD) (Saini et al., 2015). Our work is the first one to detect plant stress using SNARE proteins. In our work, we focused on extracting the features from the protein sequence with the help of convolutional neural networks, and thereby we emphasized the presence of SNARE proteins in estimating the abiotic stress in plants. The main contributions of our work are:

1. To project the involvement of SNARE proteins in alleviating the abiotic stress in plants.

2. An improvised feature extraction method to analyze the occurrence of amino acids in the protein-protein interaction of plant cells.

3. A deep neural network architecture for classifying the SNARE and non-SNARE proteins in plant protein sequences.

4. Visualization of hidden features of the network is done using manifold discovery and analysis (MDA).

Related works

Dey et al. (2022) identified the diseases that different pathogens attacked in the rice crop, leading to the manifestation of diseases such as brown spot, hispa, and NPK (nitrogen, phosphorus, and potassium) shortage. In order to precisely detect the stress in plants, the dataset was gathered from the paddy field and analyzed with several current models, including restnet50, VGG 16, VGG 19, and Inception V3.The deep CNN model performed well in terms of classification.

The work done by de Melo et al. (2022) focused on identifying water stress in sugarcane crop using thermal images as manual visualization is time-consuming. They used an integrated model called Inception-ResnetV2 which gave good accuracy with low time than traditional machine learning models. Thermal images would identify the stressed and non-stressed images by comparing the intensity distribution on the images.

Yi et al. (2020) worked on identifying the nutrient deficiency in sugarbeet crop with the help of sugarbeet nutrient deficiency dataset. They compared various CNN algorithms to find out the best model that would detect nitrogen, phosphorus, potasium and calcium deficiency. Among all the models, Denset has shown promising results. In the work done by Shona et al. (2022) cold stress in watermelon has been analyzed as this is rich in water content and nutritional fruit consumed in many countries. Here the crop is dried under a controlled environment by inducing the perfect temperature for the growth of this plant. The stress in this plant was detected based on morphological features like reduction in leaf size, shape, and leaves. U-Net architecture is used for the classification of stressed and non-stressed plants and the model achieved 100% accuracy. This article (Zhou et al., 2021) projected the importance of high throughput phenotyping in measuring crop traits. This work mainly aims to estimate the injury caused by flood currents to the soybean plant. Soya bean plants are grown under controlled conditions where the plant shows flooding symptoms and the images are collected using Ariel equipment. The flooding injury score was calculated for 724 species and five features was analyzed namely, canopy temperature, normalized difference vegetation index, canopy area, width, and length, and the model outperformed in identifying the flooding stress. This study carried out by Zhang et al. (2020) worked on identifying crop leaf purplings which usually occur when a plant is subjected to stress. Purple rapeseed leaves are examined using a UAV vehicle at pixel level with limited resolution and are segmented using U-Net architecture. The focus is to identify nitrogen stress. Leaf purpling is mainly due to a lack of nitrogen (N), phosphorus (p), and potassium (k). Accurate prediction is done with U-Net with a patch size of 256 × 256. The work (Chandel et al., 2021) focused on identifying the water stress in plants which causes huge losses to agricultural produce. The author carried out this work on maize, soybean, and okra with the help of traditional DL algorithms like Inception V3, AlexNet, and googleNet to classify and identify the plant stress. The data used for this work was collected in real-time from the crops and a total of 12,000 images were gathered from the fields using specialized cameras the convolutional neural network architectures were applied with a set of hyper-parameters and classification was done successfully in identifying the stressed and non-stressed species. Yu, Fang & Zhao (2021)’s work depicted the negative effects of heavy metal stress in tobacco plants. Unsupervised machine learning algorithms have shown remarkable results in identifying heavy metal stress. Hyperspectral images of the tobacco plant canopy were collected. Least-squares discriminant analysis (PLS-DA) and least-squares support vector machine (LS-SVM) algorithms are employed to identify heavy metal stress in plants and analyze different chemical compositions associated with it. Analysis of the nitrogen deficiency in sorghum plants has been done by Azimi, Kaur & Gandhi (2021). The author trained a CNN model to automatically detect the stress in plants. The shoot images were collected from Donald University and the proposed model achieved 96% accuracy in detecting the nitrogen stress. In the work done by Das et al. (2020), authors determined the method to identify the salt stress in the most produced and consumed food i.e., rice and it suffers from abiotic stress when adequate/low concentration of NaCl gets deposited in the soil. K-nearest neighbors (KNN) are used to classify the stressed and non-stressed and principal component analysis was used to extract features from the spectral images and partial least square regression and supervised machine learning models have performed efficiently in classifying the salt-stressed species.

Dao, He & Proctor (2021) compared the involvement of deep learning and machine learning in identifying drought stress using spectral analysis. Drought stress impacts are complicated and it affects leaves at multiple stages of severity. The derivative spectra achieved 97.5% accuracy with the help of DL algorithms. The goal of the work proposed in Zahid et al. (2022) is to discover water stress in the plant species of basil, coriander, parsley, coffee, bay leaf, and pea. With the aid of supervised machine learning techniques like random forest, SVM, and KNN, stress can be classified and predicted. By examining the morphological aspects of the data, the KNN was followed by the KNN94.64% accuracy and the SVM89.67% accuracy in terms of accuracy. Additionally, RF performed better than other classifiers in identifying water-stressed leaves and has a 99.42% accuracy rate. The work of Niu et al. (2021) identified the water stress in maize using multi-spectral images. The experiment was conducted in open maize fields in China and UAV-based RGB imagery was used to analyze the effectiveness of sensors. Random forest, ANN, and multivariant linear regression are used for the classification and identification of crop water stress and growth stages of the plant. Fractional vegetation cover (FVC) is the key measure in identifying the water stress and these models accurately calculated the FVC. Mondal et al. (2019) worked to identify the water stress in wheat. In this work, these images are captured by the Indian Agriculture Research Institute. The stressed species are classified using a random forest (RF) algorithm. Supervised machine learning techniques have shown outstanding results in analyzing the spectral data from the high-resolution images and accurate features are selected that could easily provide reliable data for analyzing the water stress in plants.

Ly et al. (2018) suggested a model to identify the environmental stress in wheat. This study mainly concentrates on identifying whether a plant is adaptable to stress or not. An extension of factorial regression called the genomic random regression model is used to predict the stress-resistant variant. Khatoon et al. (2021) has been done on analyzing the nutrient deficiency in tomato plants. The images were gathered from the field as of sufficient nutrient supply is very crucial in plant breeding. A deep neural network called DenseNet 121 has shown promising results in capturing the stressed species accurately.

The work done by Asefpour Vakilian (2020) explained the role of miRNA in regulating plant stress responses. Several machine learning algorithms like decision trees, naïve Bayes, and support vector machines are used to predict the plant stress based on miRNA concentration.

From the above works, we conclude that identifying abiotic stress in plants is found to be prominent in analyzing the productivity of the yield and several traditional methods tried to detect the cause of stress with some limitations. We herein propose a hybrid model to overcome the hurdles of the previous models in spotting and processing the important hidden features from the protein patterns. Our deep learning framework focus on extracting the reliable features from the given input patterns by incorporating traditional CNN along with RBF and to classify the sequences based on the presence of SNARE proteins we used Bi-directional LSTM for classification. Moreover, we performed comparisons with the existing methods to protect the supremacy of our proposed approach. The above literature survey is summarized in Table 1.

Table 1 Findings of various abiotic stressors for plant species.

S.No	Crop	Abiotic stressors	Dataset	Framework	
1	Rice	Phosphorus deficiency	On-field images from paddy farm	VGG19	
Potassium deficiency	
Nitrogen deficiency	
2	Sugarcane	Drought stress	Thermal images from the controlled environment	InceptionResNetV2	
3.	Sugar beet	Phosphorus deficiency	Digital images from the controlled environment	DenseNet161	
Potassium deficiency	
Nitrogen deficiency	
Calcium deficiency	
4.	Watermelon	Cold stress	Images collected under controlled environment	U-Net	
5	Soya bean	Flood stress	UAV based images	Deep CNN	
6	Purple rapeseed	Nitrogen stress	Images collected from field	U-Net	
7	maize, soybean, and okra	Water stress	Images gathered from field	Inception V3, AlexNet, and GoogleNet	
8	Tobacco	Heavy metal stress	Hyperspectral images	SVM	
9	Sorghum	Nitrogen deficiency	Shoot images	CNN	
10	Rice	Salt stress	Hyperspectral images from Paddy field	Machine laerning algorithms	
11	Wheat	Drought stress	Publicly available dataset	Deep learning algorithms	
12	Basil, coriander, parsley, coffee, bay leaf, and pea	Various stress	Public dataset	SVM and KNN	
13	Maize	Water stress	Multi spectral images using UAV from field	Random forest, ANN, and multivariant linear regression	
14	Wheat	Water stress	Images captured by Indian Agriculture Research Institute	Supervised machine learning techniques	
15	Wheat	Environmental stress	Field images	Genomic random regression model	
16	Tomato	Phosphorus deficiency	Field images	DenseNet121	
Potassium deficiency	
Nitrogen deficiency	
Calcium deficiency	

Materials and Methods

This section illustrates the sequential flow of protein-protein interaction in extracting the features from the Amino acid sequences.

Data collection

The information was taken from the NCBI-National Centre for Biotechnology Information’s Uniprot database, one of the most complete databases for protein sequences (https://www.uniprot.org/), which is in the public domain. We collected the SNARE protein data by performing search operations using the “SNARE” keyword. To handle the binary classification problem, we also collected the negative dataset with the help of the “Non-SNARE” keyword. Then we removed the redundant data with the help of the BLAST tool. Table 2 shows the statistics of the data used in our work. To extract the protein data, we collected both positive and negative data. We partitioned the data into 85% for training and the remaining 15% for testing.

Table 2 Composition of SNARE and non-SNARE protein sequences.

Types of sequence	Training	Testing	
SNARE protein sequences	4,493	1,125	
Non-SNARE protein sequences	2,569	1,078	

Pre-processing

Data collected was now pre-processed to remove the redundancy and to extract the specific format of the representation of amino acids. For this process, we used the PSI-BLAST algorithm of the NCBI repository where the protein sequences are passed as input and we extracted the position-specific scoring matrix (PSSM) by setting the parameters. This generated matrix is used to identify the hidden structures of the protein sequences.

Encoding the amino acid representation

To extract the features from the protein sequences, we calculated the PSSM for the FASTA sequences. PSSM is composed of the representation of the frequency of amino acids in the protein sequence. This is calculated by rendering the ordering of similar amino acids occurring in different sequences. For this reason, PSSM is widely used in several bioinformatics applications. The data retrieved from the Uniprot database is in the form of FASTA sequences and it is decoded into PSSM with the help of PSI-BLAST. The amino acids in a protein sequence as represented in the Fig. 1.

Figure 1 Encoding amino acids in protein sequences.

To generate this the PSI-BLAST searches for the non-redundant sequence representations by performing two iterations and the feature vector is generated represented in the form of M x 20 matrix and A = {Pa,b; a = 1……m and b = 1………20} and the PSSM is defined as follows

A={P1,1P1,2P1,3⋯⋯⋯⋯P1,20P2,1P2,2P2,3⋯⋯⋯⋯P2,20P3,1P3,2P3,3⋯⋯⋯⋯P3,20P4,1P4,2P4,3⋯⋯⋯⋯P4,20⋮⋮⋮⋮⋮Pm,1Pm,2Pm,3⋯⋯⋯⋯Pm,20}

where Pa,b represents the probability of a mutation to both amino acids during the evolution of the features from the sequential data. The amino acid protein sequences are represented in the form of a vector that holds 20 different values which symbolizes the kind of amino acid involved in them. The main function of this step is to generate a vector that can be easily modeled by CNN. Now each of these 400D vectors is normalized between 0–1 by dividing each vector by the length of the sequences. By incorporating the features of NLP, we encoded amino acid representation. The main objective is to effectively use the applications of NLP in the field of biological sequences. To extract the information from the PSSM the whole sequence should be taken as a word. For this purpose, we used fastText a model developed by Facebook that considers a word as a continuous bag of n-grams extension of word2vec (Bojanowski et al., 2017). The main idea is to treat protein sequence as a sequence and Amino acids as words and subsequently, we have generated the feature vectors of reduced dimensionality. Figure 2 represents the amino acid values.

Figure 2 Protein sequence to PSSM mapping.

Proposed work

Deep learning model

Each protein sequence has 20 amino acids and these are represented by capital letters. The deep learning model used in our work consists of a coding layer which is used to represent each amino acid in the form of a number to translate the amino acid sequence into continuous vectors we use an embedding layer and then for feature extraction we use convolution layer along with radial basis function to capture high dimensional feature vectors and to classify the sequences into SNARE and non-SNARE, we use Bi-LSTM.

The protein sequences have several features and the most common of them is the 20 amino acids with the help of several traditional feature extraction methods, Table 3 shows the common features of a protein sequence. CNN proved the best model which automatically extracts important features from a protein sequence. This is done by our hidden layers of the network which contain various parameters and shapes that determine the quality of our model.

Table 3 Features of protein sequence.

S.no	Features	
1	A-Alanine	
2	R-Arginine	
3	N-Asparagine	
4	D-Aspartate	
5	C-Cysteine	
6	Q-Glutamine	
7	E-Glutamate	
8	G-Glycine	
9	H-Histidine	
10	I-Isoleucine	
11	L-Leucine	
12	K-Lysine	
13	M-Methionine	
14	F-Phenylalanine	
15	P-Proline	
16	T-Threonine	
17	S-Serine	
18	W-Tryptophan	
19	Y-Tyrosine	
20	V-Valine	
21	CCD-Coiled-coil domain	
22	A-Nitrosyaltion (Nito A)	
23	B-Nitrosyaltion (Nito B)	
24	C-Nitrosyaltion (Nito C)	
25	Total Nitrosyaltion (Total Nitro)	
26	A-Nitrityrosine (YNO A)	
27	B-Nitrityrosine (YNO B)	
28	C-Nitrityrosine (YNO C)	
29	Total Nitrityrosine (YNO Total)	
30	SUMOylation I (SUMO I)	
31	SUMOylation II (SUMO II)	
32	SUMOylation III (SUMO III)	
33	TotalSUMOylation (SUMO Total)	
34	Amio acid number	
35	Number of negative amino acids	
36	Number of positive amino acids	
37	Molecular weight	
38	Theoretical PI	
39	Total carbon atoms	
40	Total hydrogen atoms	
41	Total nitrogen atoms	
42	Total oxygen atoms	
43	Total sulphur atoms	
44	Instability index	
45	Aliphatic index	

To enhance the performance results of the model and to prevent overfitting we have used dropouts where the model randomly deactivates the neurons and the computational time is reduced by tuning the dropouts and the non-linear function called rectified linear unit (ReLU) was applied after each operation as in Eq. (1)

(1) f(x)=max(0,x)

The model was then evaluated with a softmax activation function which explains the output with the help of a logistic function as shown in Eq. (2).

(2) σ(Z)i=ezi∑k=1k⁡ezk

where the input vector for the k-dimensional vector is represented as Z in a range of values (0, 1).

CNN-RBF

The radial basis function has shown promising results in the evaluation of features from the protein sequences related to abiotic stress in plants. In our work, we used RBF’s functionality to establish the relationship between amino acid composition and abiotic stress responses in plants as it can capture non-linear relationships between the data. The radial basis function is typically used as an activation function in the network. The output of the node is calculated by measuring the distance between the center of the node and the input, RBF works accordingly. The hidden layer performs linear regression to anticipate the outputs by performing non-linear transformations of input and output layers as RBF can have multiple hidden layers at a time in an active state. The Gaussian radial basis function is generally expressed in the below equation Eq. (3)

(3) φ(s)=exp(−γ∗s2)

where φ(s) represents the output of the radial basis function γ is the parameter that minimizes the spread of width of the function and s is the distance calculated between the center of the node to the input. By adjusting the RBF node distance values and γ, the network can effectively establish a relationship between the protein sequence amino acid representations and abiotic stress responses. Feature identification is an important part of the analysis and CNNs have shown remarkable results in capturing the local patterns and hierarchical features on the other hand RBF networks work well with handling the nonlinear relationship between the features. By incorporating the RBF component into the CNN model, the hybrid architecture captures the local and global features of the protein sequences and establishes complex non-linear relationships between them which play a key role in protein structure analysis. The integrated outputs of both RBF and CNN modules are combined to extract the hierarchical features. The embedding obtained from a fully connected layer of CNN is flattened and fed into RBF. RBF kernels are non-linear and increase the complexity whereas when integrated with CNN as the backbone, a simple linear quadratic activation function in the space of r2 which is calculated by squaring the cluster centers with samples is defined as follows in Eq. (4)

(4) p(r)=1−r2/σ

where σ defines the width of the kernel. The overall architecture of CNN with RBF as a classifier for high-dimensional feature representation is shown in the below figure.

Bi-directional long short-memory

To model the protein sequences to learn the long-range dependencies of the sequential data and to identify the presence of SNARE proteins in a sequence we applied bi-directional long short-memory (Bi-LSTM) which updates the hidden states for sequential data from two directions. The architecture works as shown in Fig. 3 the Bi-LSTM layer consists of two layers of LSTM where one layer receives the input in the backward direction (hpt) and the other layer receives the input in the forward direction (hft) where t = (1, 2, 3, ….. n). The output is given by combining the inputs of forward and backward layers. The Bi-LSTM layers compute Hid=(h1,⋯⋯,ht) and Out=(o1,⋯⋯,ot) output obtained from hidden layer and output layer as in Eqs. (5) and (6),

Figure 3 Structural overview of Bi-LSTM.

(5) Ht=ACT(Wshht+Whhht−1+Biah)

(6) Ot=Whoht+Biao

where Wsh is the weight matrix between the input layer and the intermediate layer Biah is the bias vector computed for the intermediated layer and ACT is the non-linear vector activation function. To calculate the presence of SNARE proteins in the given seq we use the function F(seq) as shown in Eq. (7)

(7) F(seq)=Bi−LSTM(CNN+RBF(Embedding(Encoding(seq)).

Feature fusion model

We proposed a feature fusion model that combines the functionalities of CNN, RBF, and Bi-LSTM to detect abiotic stress in plants. The architecture of the model is shown in Fig. 4. This hybrid approach focuses on projecting the hierarchical feature extraction capabilities of CNN non-non-linear modeling of RBF networks and the sequential analysis of Bi-LSTM. This hybrid approach influences the strengths of different neural network architectures to potentially improve the overall performance and capture complex patterns in the data. The main contribution of the feature fusion model to our sequential data is due to the following attributes that contribute to identifying abiotic stress in plants effectively:

Figure 4 Hybrid model architecture.

1. Hierarchical feature extraction:

CNNs are excellent at learning hierarchical features from raw data, such as images or sequences. The significant property is that they can capture low-level features and gradually learn more abstract features. When the CNNs are combined with subsequent layers like RBF and Bi-LSTM, the model is capable of capturing both local and global patterns in the data.

2. Nonlinear and radial basis function (RBF) modeling:

RBF networks are good at approximating complex nonlinear relationships. They can learn to map input features to nonlinear functions, which might be particularly useful for capturing intricate patterns that CNNs alone may not fully grasp.

3. Enhanced representation learning:

Through the deployment of the fusion model, there is a greater possibility of enrichment in feature representations of the data. CNNs can capture spatial or sequential patterns, RBF networks can capture complex nonlinear relationships, and Bi-LSTMs can handle sequential dependencies. These models together would be able to learn more informative and discriminative representations.

4. Better generalization:

Through this combination, the model can generalize better to unseen data by utilizing diverse approaches to capture various patterns.

Combining all the aforementioned benefits, our Feature fusion model forms a robust and resilient framework that magnifies the functionalities of these components in identifying the abiotic stress in plants based on the biological sequential pattern analysis of amino acids. This approach surpasses the limitations of the traditional methods by ensuring an elevated outcome.

Algorithm explanation

Input sequences denoted as P are collected and then a PSSM is generated with dimensions 20 × 20 and then A represents the PSSM. Aaug denotes the Augmented matrix and then the original and augmented PSSM is denoted as Pcom. Fcnn(Pcom) denotes the embedding vectors generated by CNN. Where fij represents the jth feature value obtained by applying the ith filter to the combined sequence. μ represents a set of RBF centers and RBF(μ,Fcnn(Pcom)) represents the combined feature embedding vectors obtained from cnn and rbf Where rbfij represents the RBF activation value for the ith CNN feature vector and the jth RBF center. Let Fcom(P) represent the combined feature vectors from the CNN and RBF layers. h_t(f) be the hidden state at time step t for the forward LSTM and ht(b) be the hidden state at time step t for the backward LSTM, ot is the output vector at time step t. Wo is the weight matrix connecting the hidden states to the output layer. The concatenated hidden state vector at timestamp t (both forward and backward hidden states) is represented as Ht. bo is the bias vector associated with the output layer.

Algorithm Algorithm for hybrid feature fusion model.

Input: Protein sequence P [seq1, seq2, seq3, ……….. seqn]	
Output: Stress classification [Stressed, non-stressed]	
Step-1: Process the Protein sequence and generate PSSM as P[MX20]	
Pa,b:a=1⋯mandb=1⋯20

A=[Pm1,Pm2,⋯⋯⋯Pm20]

	
Step 2: Augment the generated PSSM as	
Aaug=A[Pm1,Pm2,⋯⋯Pm20]+P|[Pm1|,Pm2|⋯⋯Pm20|]

Pcom=A[Pm1,Pm2,⋯⋯⋯Pm20,Pm1|,Pm2|⋯⋯Pm20|]

	
Step 3: Generate Feature Embedding Vectors	
Fcnn(Pcom)=[(f11,f12,⋯⋯f1m),(f21,f22,⋯⋯f2m),⋯⋯(fn1,fn2,⋯⋯fnm)]RBF(μ,Fcnn(Pcom))=[(rbf11,rbf12,⋯rbf1k),(rbf21,rbf22,⋯rbf2k),⋯(rbfn1,rbfn2,⋯rbfnk)]

	
Step-4: Classification of stressed and non-stressed	
for each timestamp from 1 to T:	
ht(f)=LSTMforward(Fcom(P)t,h(t−1)(f)) #forward LSTM	
for each timestamp from T to 1:	
ht(b)=LSTMbackward(Fcom(P)t,h(t−1)(b)) #backward LSTM	
Combine Hidden states:	
for each timestamp t from 1 to T:	
Ht=[ht(f);ht(b)]

	
Output:	
Ot=softmax(wo∗Ht+bo)

	

Performance evaluations

This study was aimed at identifying the presence of the SNARE protein in the protein sequence and for this, we need a positive dataset to represent SNARE-protein sequences and a negative dataset to represent non-SNARE protein sequences. For the dataset, hyper-parameters are employed to identify the best model. The efficiency of the model is evaluated as accuracy, recall, specificity, F1-score and Matthews correlation coefficient (MCC) using true positive (TP) indicies. In confusion matrix, additional data is represented using true negative (TN) values. The incorrect values are analysed using false negative (FN) and false positive (FP) values. The evaluation metrics MCC is mentioned in Eq. (8).

(8) Matthewscorrelationcoefficient(MCC)=(TN∗TP)−(FN∗FP)((FP+TP)(FN+TP)(FP+TN)(FN+TN))0.5

Results

The reliability and quantitativeness of our model are evaluated by evaluating the data with other architectures. The evaluation results and comparisons of different architectures are enclosed in this part.

Distribution of occurrence of the amino acids

Our work analyzed the composition of amino acids in SNARE and non-SNARE sequences by computing the frequency of occurrence of amino acids in each sequence is shown in Fig. 5.

Figure 5 Amino acid distribution analysis.

Comparative analysis in identifying SNARE protein with CNN and other networks

As SNARE proteins constitute a fraction of the occurrence of abiotic stress in plants, we examined the input data by applying several algorithms like CNN, LSTM, CNN with Bi-LSTM, and CNN with RNN to evaluate the performance of these models with our CNN model. Table 4 shows the comparative analysis. Our model showed remarkable performance when compared with other machine learning algorithms. The hyperparameters of the model are shown in Table 5.

Table 4 Comparative analysis of various model.

Classifier	Accuracy	Sensitivity	Specificity	MCC	
CNN	65.1	80.4	55.7	0.4	
CNN+RNN	68.6	75.5	66.1	0.2	
CNN+Bi-LSTM	69.4	81.3	67.8	0.3	
LSTM	70.1	82.6	58.5	0.1	
Proposed model	74.6	88.8	73.1	0.4	

Table 5 Model hyperparameters.

Hyper parameters	Values	
Epochs	50	
Batch size	32	
Learning rate	0.001	
Optimizer	Adam	

The composition of the protein sequences is visualized by calculating the similarity distance between them and the same is shown in Fig. 6. The compositions of the input patterns involved in identifying the abiotic stress in plants are calculated by observing the proportion of the SNARE proteins involved and the Fig. 7 shows the distribution of the input data.

Figure 6 Comparison of sequence similarity between SNARE and non-SNARE proteins.

Figure 7 SNARE and non-SNARE protein composition comparison.

To evaluate the correctness of the model in classifying the stressed and non-stressed input patterns is represented in the form of a confusion matrix as shown in Fig. 8. A graphical representation to predict the performance of the model on various thresholds to distinguish and classify the positive input data and the negative input data is shown in Fig. 9. The consolidated analysis of the performance metrics used to evaluate the classification of the stressed and non-stressed input data is shown in Fig. 10.

Figure 8 Confusion matrix.

Figure 9 Receiver operating characteristic curve.

Figure 10 Assessment of performance metrics for input data.

Visualization and analysis of feature space

Our study employs various visualization techniques to enhance our understanding of the feature space for identifying abiotic stress in plants using SNARE protein sequences. We utilize t-SNE (t-distributed Stochastic Neighbor Embedding), manifold Discovery and analysis (MDA), assessments of noise reduction, and examinations of intermediate layers to comprehensively analyze our model’s efficacy and interpretive aspects across multiple dimensions. The t-SNE visualization of input data is shown in Fig. 11.

Figure 11 Visualization of features using t-SNE.

MDA visualization of feature space

A brief description has been provided in the Supplemental File. As feature extraction plays a key role in identifying the roots of the plant stress, deep neural network outperforms the traditional methods by extracting millions of features related to the task. To attain the topological information and to preserve the local geometry of the feature space, MDA showed remarkable results in providing insightful visualization of neural network features by reducing the dimensionality of intermediate layer features to 2D, facilitating visualization via scatter plots (Islam et al., 2023). In our study, To further explore our network’s behavior, we analyzed the distribution of data in one of the final layers across different epochs, depicted in Supplemental Files S1–S3. This visualization helps understand how the network's feature extraction process evolves over training cycles. MDA also enables the visualization of training data over epochs, projecting various coverage patterns, as shown in Supplemental Files S4–S6. This analysis provides valuable insights into the network's training dynamics. To enhance our model's robustness, we introduced Gaussian noise to the test data and visualized the feature distribution using MDA. Supplemental Files S7 and S8 demonstrates the visualization of trained data and, highlighting how the network’s performance is affected by Gaussian noise. Finally, we compared the MDA visualization of our proposed Hybrid model with DenseNet in Supplemental Files S9 and S10, showcasing the differences in feature extraction and classification performance between the two models. These figures are provided in the Supplemental Files.

Conclusion

Predicting abiotic stress in plants is essential in today’s scenario to safeguard agricultural productivity. SNARE proteins play a key role in regulating the metabolism of the amino acids that lead to the occurrence of abiotic stress in plants. Identification of SNARE proteins is of major concern in biological computation. SNARE proteins play a key role in regulating the metabolic activity of the cellular structure; hence, it is essential to develop models to identify their occurrence in the protein sequences. In our study, we proposed a hybrid model by replacing the shortcomings of traditional machine learning algorithms used to identify hidden patterns in amino acid sequences. In our approach, we extracted the features from the sequences with the help of a CNN, and then with the help of a radial basis neural network, we generated the high-dimensional feature vectors. To detect the stressed and non-stressed sequences we applied bi-directional LSTM to classify the sequences. We applied many experiments to validate the performance of our approach with the existing models. The experimental results obtained using our approach surpassed the existing methods. To our knowledge, this is the first approach used to identify the presence of SNARE proteins that detect the occurrence of abiotic stress in plants. Furthermore, our approach may facilitate the discovery of underlying functionalities of different proteins in the future by revealing the hidden structure of the sequences with the help of feature extraction methods.

Supplemental Information

Supplemental Information 1 Input vector.

Supplemental Information 2 Data distribution in Intermediate layer-1.

Supplemental Information 3 Data distribution in Intermediate layer-2.

Supplemental Information 4 Data distribution in Intermediate layer-3.

Supplemental Information 5 Data Distribution at Epoch-1.

Supplemental Information 6 Data Distribution at Epoch-5.

Supplemental Information 7 Data Distribution at Epoch-10.

Supplemental Information 8 Feature Distribution Visualization of Trained data.

Supplemental Information 9 Feature Distribution Visualization of Test data by adding Gaussian noise.

Supplemental Information 10 Feature Space Analysis of Hybrid Model Using MDA Visualization.

Supplemental Information 11 Feature Space Analysis of DenseNet Using MDA Visualization.

Supplemental Information 12 Rice Fasta sequence data.

Supplemental Information 13 Fasta sequences of snare in Rice.

Supplemental Information 14 Fasta sequences of Non-snare in Rice.

Supplemental Information 15 The code related to abiotic stress in plants using snare proteins.

This is a hybrid feature fusion model where high-level features are extracted by integrating the RBF neural network with CNN and the extracted feature maps are classified into stressed and non-stressed using Bi-lstm model. The features are analyzed using MDA visualization techniques.

Supplemental Information 16 Code for identification of plant stress.

Additional Information and Declarations

Competing Interests

Author Contributions

Data Availability

The authors declare that they have no competing interests.

Bhargavi T. conceived and designed the experiments, performed the experiments, performed the computation work, prepared figures and/or tables, and approved the final draft.

Sumathi D. analyzed the data, authored or reviewed drafts of the article, and approved the final draft.

The following information was supplied regarding data availability:

The raw data and code are available in the Supplemental Files.

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
