# Peer review of "Early detection of abiotic stress in plants through SNARE proteins using hybrid feature fusion model"

_PeerJ Computer Science, doi:10.7717/peerj-cs.2149_

## Round 0.1 · original submission · Major Revisions

The article must be improved according to all suggestions from reviewers!

**Language Note:** PeerJ staff have identified that the English language needs to be improved. When you prepare your next revision, please either (i) have a colleague who is proficient in English and familiar with the subject matter review your manuscript, or (ii) contact a professional editing service to review your manuscript. PeerJ can provide language editing services - you can contact us at [email protected] for pricing (be sure to provide your manuscript number and title). – PeerJ Staff

·

Basic reporting

The authors have used some old references in related work and the introduction section. Please go through some recent state-of-the-art published work related to the field and update it accordingly.
• The related work lacks coherence. To address this, consider presenting the information in a tabular format, allowing for a comparison of methods based on criteria such as technique, dataset, novelty, etc.

Experimental design

Can you provide more details regarding the concept of "early detection" mentioned in the manuscript's title? Does it pertain to the identification of abiotic or biotic stressors before their visible symptoms become apparent? If this is the case, kindly elaborate further on the process of early detection.

• Does Processing the protein information in 20X20 dimension can cause loss of data? Is their any way to process it in 1D form?
• Please mention the train test and validation split in Figure 4.
• Additionally, include details about the hyperparameters utilized during the model training process.

Validity of the findings

mentioned above

Reviewer 2 ·

Basic reporting

No comments

Experimental design

Further benchmarking is necessary with state-of-the-art tabular data analysis techniques such as DeepInsight [1] and Genomap [2].

[1] Sharma, A., Vans, E., Shigemizu, D. et al. DeepInsight: A methodology to transform a non-image data to an image for convolution neural network architecture. Sci Rep 9, 11399 (2019).
[2] Islam, M.T., Xing, L. Cartography of Genomic Interactions Enables Deep Analysis of Single-Cell Expression Data. Nat Commun 14, 679 (2023).

Validity of the findings

To better validate the findings and improve the interpretability, visualization of the deep learning feature space is necessary. I would suggest the authors use deep learning feature space analysis tools such as MDA (https://github.com/xinglab-ai/mda, published in Nature Communications) to show the distribution of the data in the feature space. The authors should show a) the distribution in at least 4 intermediate layers to show how their network is performing prediction, b) the distribution of data at one layer (preferably one of the final layers) with change in epoch to better understand the feature extraction process by their network. c) add some Gaussian noise to testing data and then show the feature distribution using MDA to visualize the robustness of their network. The authors can choose any other network such as GoogleNet/DenseNet to show side-by-side comparisons of the MDA visualizations of the feature space with their network.

Additional comments

The paper is overall well-written. However, the figure resolution should be improved. The captions of the figures should be more descriptive.

---

## Round 0.2 · Major Revisions

The authors must respond more detailed to the second reviewer!

·

Basic reporting

The authors have substantially improved the manuscript and it can be accepted for publication.

Experimental design

no comment

Validity of the findings

no comment

Reviewer 2 ·

Basic reporting

For my first comment, the authors responded, "Exploration of diversified methods for analysis of data is truly required as per the suggestions. As the current focus and this work are mainly focused on the detection of the abiotic stress with the snareproteins, the authors can consider this and carry out as a further work. "

This is a very poor response and the authors should at least discuss the methods in the introduction or discussion section.

The authors used a method called MDA to create feature visualization without referring to the paper or the code repository.

Experimental design

no comment

Validity of the findings

no comment

Additional comments

no comment

---

## Round 0.3 · accepted · Accept

The paper was well improved and both reviewers accepted the paper.

Reviewer 2 ·

Basic reporting

No comments

Experimental design

No comments

Validity of the findings

No comments

Additional comments

No comments